# Comparative Analysis of Deep Neural Networks and Graph Convolutional Networks for Road Surface Condition Prediction

**Saroch Boonsiripant** [1], **Chuthathip Athan** [2], **Krit Jedwanna** [3,*], **Ponlathep Lertworawanich** [4] **and Auckpath Sawangsuriya** [4]

1    Department of Civil Engineering, Faculty of Engineering, Kasetsart University, Bangkok 10900, Thailand; saroch.b@ku.th
2    Mobinary Company Limited, Bangkok 10400, Thailand; chuthathip@mobinary.org
3    Department of Civil Engineering, Faculty of Engineering, Rajamangala University of Technology Phra Nakhon, Bangkok 10300, Thailand
4    Department of Highways, Bureau of Road Research and Development, Bangkok 10150, Thailand; ponlathep@gmail.com (P.L.); sawangsuriya@gmail.com (A.S.)
*    Correspondence: krit.j@rmutp.ac.th; Tel.: +66-89-777-1654

**Abstract:** Road maintenance is essential for supporting road safety and user comfort. Developing predictive models for road surface conditions enables highway agencies to optimize maintenance planning and strategies. The international roughness index (IRI) is widely used as a standard for evaluating road surface quality. This study compares the performance of deep neural networks (DNNs) and graph convolutional networks (GCNs) in predicting IRI values. A unique aspect of this research is the inclusion of additional predictor features, such as the type and timing of recent roadwork, hypothesized to affect IRI values. Findings indicate that, overall, the DNN model performs similarly to the GCN model across the entire highway network. Given the predominantly linear structure of national highways and their limited connectivity, the dataset exhibits a low beta index, ranging from 0.5 to 0.75. Additionally, gaps in IRI data collection and discontinuities in certain highway segments present challenges for modeling spatial dependencies. The performance of DNN and GCN models was assessed across the network, with results indicating that DNN outperforms GCN when highway networks are sparsely connected. This research underscores the suitability of DNN for low-connectivity networks like highways, while also highlighting the potential of GCNs in more densely connected settings.

**Keywords:** international roughness index (IRI); graph convolutional network (GCN); deep neural network (DNN); machine learning

## 1. Introduction

Typically, ride quality is linked to how comfortable road users feel while driving, which is influenced by the level of roughness of the pavement surface. Pavement roughness refers to the surface irregularities of the pavement; it impacts both the condition of the road and the comfort experienced by users. An escalation in pavement roughness results in higher fuel consumption, increased vehicle maintenance and repair expenses, elevated greenhouse gas emissions, and potentially increased traffic safety; overall, these factors can result in substantial financial losses annually [1].

The international roughness index (IRI) was developed by the World Bank during the 1980s. It is defined as the cumulative vertical motion of the suspension system divided by the distance traveled, derived from a mathematical model simulating a quarter-car (consisting of a wheel, one-quarter of the vehicle's body mass, and the associated suspension) moving along a measured road surface at a speed of 80 km/h [2]. Numerous highway agencies worldwide use an initial measurement of the IRI following construction as a quality assurance benchmark, while the terminal IRI serves as an indicator for necessary

pavement maintenance measures or potential reconstruction requirements. Some highway agencies have embraced the use of the present serviceability rating or present serviceability index for assessing pavement conditions. Therefore, several agencies have considered roughness as a serviceability measurement over time. Due to the crucial role the IRI plays as an indicator of pavement performance, considerable research has been directed toward modeling and forecasting IRI [1,3].

The majority of the previous IRI models were based on linear or nonlinear regression techniques. Some recent models have utilized the deep neural network (DNN) method, which is one of the machine learning algorithms [4–6]. DNNs are one type of machine learning technique, with their concept being biologically inspired by the human brain; thus, they mimic brain behavior [7]. DNNs offer highly precise solutions for constructing empirical models having complex datasets that exhibit nonlinear behavior and do not conform to known mathematical functions [8].

In general, DNNs comprise an input layer, an output layer, and multiple hidden layers, where intricate nonlinear operations are performed. Every layer comprises a group of neurons, interconnected through synapses, with initial weights that evolve during the network's iterative process. Given that DNNs handle data that does not adhere to a straightforward mathematical relationship, the ultimate solution is often regarded as a black box [4]. Abd El-Hakim and El-Badawy [5] applied DNNs to create a neural network model for predicting IRI in rigid pavements. Sollazzo et al. [6] established a correlation between pavement roughness and structural performance through the application of DNNs.

Recently, advanced deep neural network methods have been widely used in traffic prediction and have achieved good performance. Lv et al. [9] used stacked autoencoders to extract spatial–temporal traffic flow features and make traffic flow predictions. Fu et al. [10] applied the LSTM model and its variant GRU model to predict short-term traffic flow [11]. Chen et al. used convolutional neural networks (CNNs) to predict traffic flow based on time series folding and multi-grained learning techniques [12]. Yu et al. proposed a graph convolutional network (GCN) model to tackle the traffic prediction problem [13]. Li et al. [14] also used a GCN to extract spatial relationships of traffic flow between observation locations. Bai et al. [15] and Sharma et al. [16] utilized GCNs for traffic data forecasting and estimation.

Traffic forecasting for a complete network has always been a challenge due to the complex spatial and temporal correlations. With spatial correlation, variations in traffic volume are primarily influenced by the topological layout of the urban road network, with the traffic conditions on upstream roads affecting those downstream through a transfer effect, while the conditions downstream impact those upstream through a feedback effect [17]. With temporal correlation, the traffic volume fluctuates dynamically over time, primarily manifesting as periodic patterns and trends [11]. Despite numerous studies addressing spatial and temporal dependencies in traffic data, such as speed, flow, and density, there has been relatively little research on incorporating spatial and temporal dependencies into modeling using the IRI.

Therefore, this study aims to develop an IRI prediction model using the graph convolutional network (GCN) algorithm, which considers the spatial and temporal correlations of asphalt pavement conditions. Additionally, the performance of the GCN model was compared with that of a deep neural network (DNN). This comparative analysis will help determine the effectiveness of GCN in modeling spatial relationships within road networks compared to DNN.

## 2. Literature Review

### 2.1. IRI Model Features

The selection of features used in modeling is crucial. Several studies have been conducted to predict the IRI index using pavement distress, the structural number of the pavement, moisture content, climate data, and traffic data [18,19]. However, when data have occasionally not been collected immediately after roadwork maintenance, the inspec-

tion date and duration time since the last maintenance might affect the IRI index. Other studies have reported less research considering the inspection date in their IRI modeling, although they did include the age of the pavement since its construction occurred [20,21]. Therefore, the current study assigned the inspection date and duration since the last maintenance as inputs in developing the IRI model (given that the age of the pavement in the current was not available) and compared the influence of these attributes on the IRI. Furthermore, some of the road sections were subjected to different roadworks at various stages of pavement aging, with each type of roadwork having a different influence on the reduction of the IRI [22]. Hence, it was crucial to assess the individual effect of each roadwork type on the IRI.

### 2.2. IRI Prediction Models

Over the years, numerous researchers have proposed several roughness prediction models, in terms of the IRI. These models can be categorized into three groups: deterministic, probabilistic, and machine learning-based (such as deep learning and artificial neural networks) [21].

The recent developments are more inclined toward machine learning models, while the deterministic and probabilistic models are referred to as basic models that have gathered much attention [23,24]. Paterson [25] developed a linear regression model to estimate the IRI based on factors such as cracking, rutting, pavement age, structure number, equivalent single axle loads (ESAL), thickness of cracked layers, and the number of potholes. Meanwhile, Chandra et al. [26] utilized linear regression, nonlinear regression, and neural networks (NN) to predict IRI using measurements of rutting, cracking, potholes, patches, and raveling. Tamagusko and Ferreira [27] explore how machine learning (ML) is applied to predict the IRI. Sigdel et al. [28] develop models to predict the IRI for Nepal's national highways, essential for effective road maintenance. El-Hakim et al. [29] and Alnaqb et al. [30] highlight the potential of ML models to enhance IRI predictions and support sustainable and efficient pavement management.

Alternatively, cutting-edge machine learning models can be deployed that may include the dependence of pavement evaluation metrics on other parameters. This information can be utilized to predict the IRI of roads in the future by analyzing the data gathered from the road. The implementation of the 'You Only Look Once (YOLO)' approach on Indian roads has demonstrated promising outcomes in identifying potholes using deep learning methods [31].

Another compelling type of prediction model is those based on GCNs. A GCN is a category of neural network designed to work with graph-structured data. It is particularly effective for analyzing data represented as graphs, where nodes denote entities and edges signify relationships between them. However, this model has not been utilized widely for the prediction of road pavement performance [32]. Therefore, the current aimed to enhance the traditional IRI prediction model by utilizing a GCN that incorporates spatial relations from neighboring sections. Additionally, several factors impacted the IRI model. The following paragraphs will describe the importance of each of these parts and how they have influenced published research.

## 3. Data Collection

### 3.1. Data Description

The current study utilized data from the Department of Highways, Thailand, representing the overall pavement conditions of the national highways. The data were organized into 3023 highway sections, covering approximately 104,606 km in both directions. On average, the length of each section was 34.6 km. There were 2650 sections with asphalt pavement, with their associated IRI values and covering approximately 49,275 km, as shown in Figure 1.

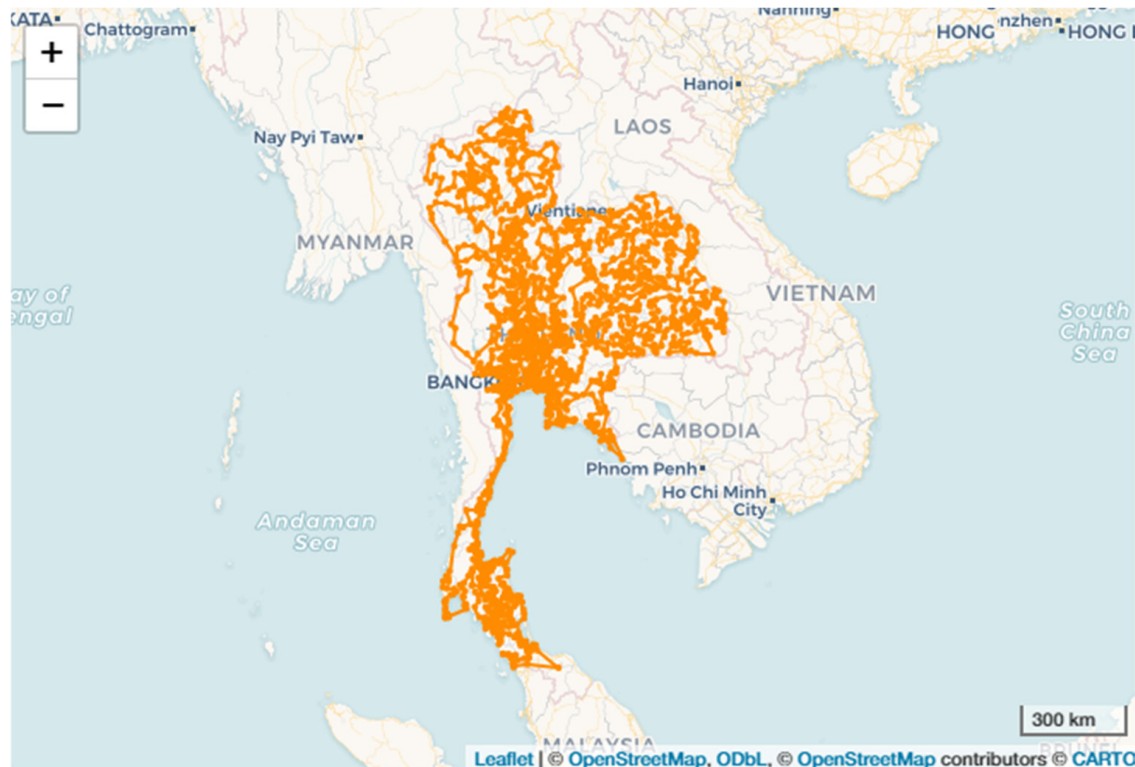

**Figure 1.** Geolocations of all asphaltic surface highway sections considered in this study in Thailand, indicated by orange lines.

These data were collected annually using pavement inspection equipment called a transverse profile logger and the laser crack measurement system, accounting for approximately 30% of the total extent of the national highways. The average IRI value in the entire network was approximately 2.45 m/km with a range of 0.25–9.50. The box plots in Figure 2 illustrate the IRI values for six road hierarchies: national highways, regional highways, provincial highways, district highways, motorways, and connectivity links. The values shown in Figure 3 are the maximum, minimum, and median for each road hierarchy, respectively.

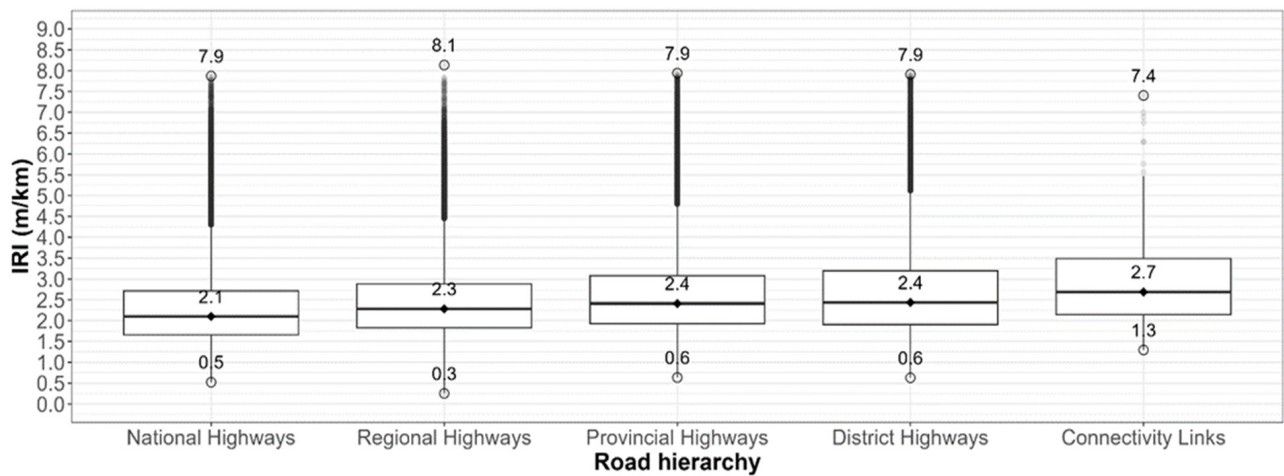

**Figure 2.** IRI data are grouped by road hierarchy.

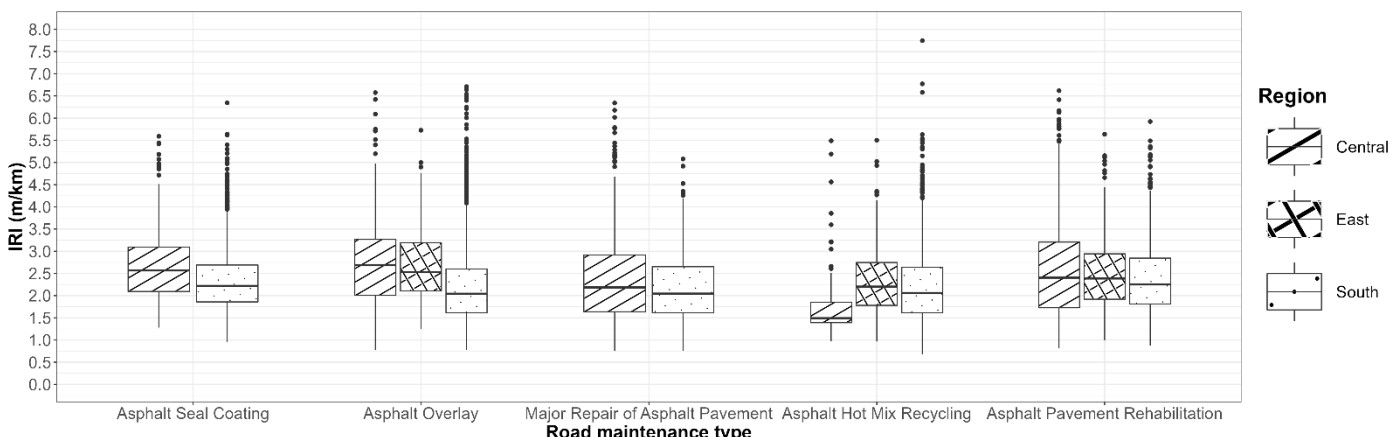

**Figure 3.** The road work distribution in each region.

The dataset included information on surveyed pavement data that had been inspected, as well as road geometries, such as IRI, pavement distress, surface type, and road hierarchy. Additionally, supplementary data were obtained: minimum temperature, maximum temperature, and precipitation from the Thai Meteorological Department; annual average daily traffic (AADT) from the Department of Highways; and the roadwork maintenance schedule log.

### 3.2. Road Maintenance Data

The IRI values were examined that had been obtained from the regular pavement performance surveys conducted on the road network in Thailand. These surveys were carried out at specific times following road maintenance activities. However, the data collection in this research spanned only four years (2019–2022, inclusive), resulting in a low number of road maintenance activities and the distributed road work data in each type and region, as shown in Figure 3. There are five types of road maintenance that are located on asphalt pavement: asphalt seal coating (22100), asphalt overlay (22200), major repair of asphalt pavement (23200), which involves addressing damage to the base, subbase, or subgrade layers by removing the deteriorated material and replacing it with new material, followed by resurfacing, asphalt hot mix recycling (23300), and rehabilitation of asphalt pavement (24100). The total number of road maintenance records amounts to 10,859 records over the four years, which is shown in Figure 4.

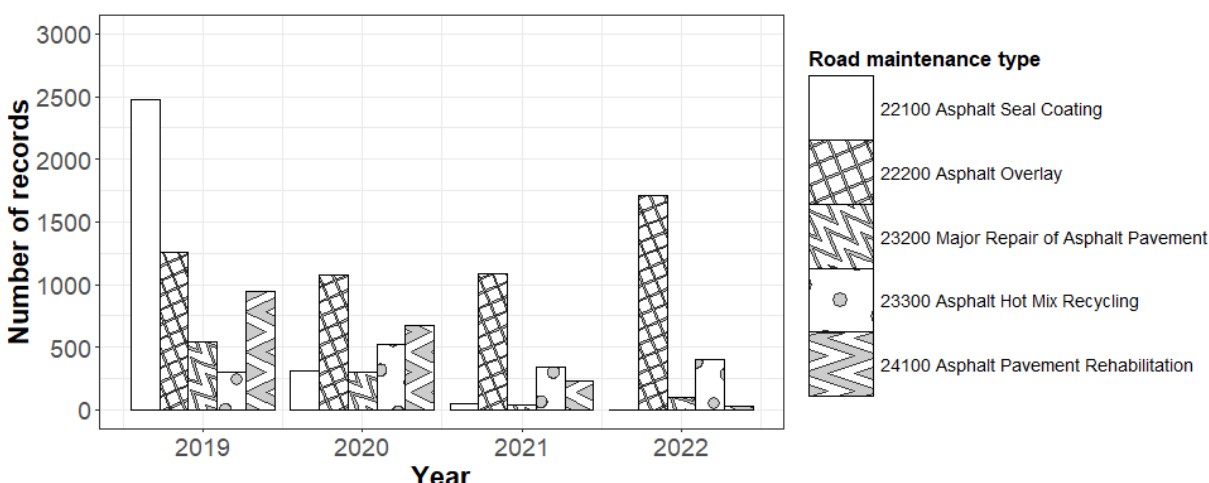

**Figure 4.** The number of road maintenance records each year.

## 4. Data Preparation

This section explains how the traffic data and other parameters were handled prior to their use in the model development. There were four steps in processing these data, with the details of each step described in the following sections.

### 4.1. Road Segmentation

The sections were divided into 100 m segments, resulting in 1,087,989 segments with IRI values for the asphalt pavement, spanning the period of 2019–2022.

### 4.2. Data Cleaning

Before using the data to develop the IRI model, it was important to properly handle anomalous data. Specifically, unreasonable cases were identified, such as instances where the IRI increased in sections undergoing road maintenance or decreased in sections without road maintenance. These issues arose because certain maintenance activities were not recorded in the database; thus, these records were removed before utilizing the data in model development. Additionally, sections that lacked IRI data for 2022 (the year used as the target value) or had data only for 2022 but not for other years were also removed. Approximately 30% of these data were removed during the data cleaning phase.

### 4.3. Attribute Coding

The dataset used in this study included various numerical, binary, and categorical attributes relevant to predicting the international roughness index (IRI). The IRI_[year] attribute served as both a feature and target variable, with values from 2019 to 2021 used as features and 2022 as the target. The data were split into training (80%) and test (20%) sets, ensuring equal representation by beta index, region, and an 80/20 split based on total distance. Table 1 provides a summary of key data attributes, including indicators of road distress and climate variables like monthly temperature (m[01]temp_min to m[12]temp_max) and monthly precipitation (m[01]rain_total to m[12]rain_total) to capture weather effects. Categorical variables were converted into dummy variables, such as PLAN_CODE[ID], to indicate maintenance activities and road hierarchy, while binary variables like IND_IRI[year] and IND_AC[distress type][year] flagged missing IRI or road distress data for each year. RW_INSP_[year] indicates whether maintenance occurred before the IRI inspection.

**Table 1.** Data attributes.

| Feature | Description | Unit | Feature Type |
|---|---|---|---|
| IRI_[year] | IRI | m/km | Numeric |
| m[01]_temp_min m[12]_temp_min | Minimum temperature in each month | °C | Numeric |
| m[01]_temp_max m[12]_temp_max | Maximum temperature in each month | °C | Numeric |
| m[01]_rain_total m[12]_rain_total | Total precipitation in each month | mm | Numeric |
| PLAN_CODE_[ID] | Road maintenance ID | - | Categorical |
| IND_IRI_[year] | Missing data indicator for IRI | - | Binary |
| IND_AC_[distress type]_[year] | Missing data indicator for road distress | - | Binary |
| HIER_[number] | Road hierarchy | - | Categorical |
| RW_INSP_[year] | Indicator for road maintenance occurring before IRI inspection | - | Binary |
| AADT_[year] | Annual average daily traffic | veh/day | Numeric |
| TRUCK_[year] | Proportion of trucks | % | Numeric |
| NUM_INSP_END_ [year] | Number of days from inspection date to end of year | day | Numeric |
| NUM_INSP_RW_ [year] | Number of days from latest day of inspection to earliest road maintenance | day | Numeric |

Additional features included HIER_[number] for road hierarchy, AADT_[year] for annual traffic volume, and TRUCK_[year] for the proportion of trucks, which are crucial for understanding pavement wear. Maintenance-related features, such as RW_INSP_[year] and NUM_INSP_END_[year], captured the timing of maintenance and inspections, enabling an in-depth analysis of spatial, temporal, and maintenance influences on IRI.

### 4.4. Data Normalization

Data normalization is required because various types of data used for model predictions have differing magnitudes and units. This process transforms the data into a more consistent and standardized range. To achieve this, the means and standard deviations were computed from the training dataset and applied to Equations (1) and (2) for both the training and test datasets, resulting in a normalized dataset.

$$z_{train} = \frac{x_{train} - \mu_{train}}{\sigma_{train}} \tag{1}$$

$$z_{test} = \frac{x_{test} - \mu_{train}}{\sigma_{train}} \tag{2}$$

where $z$ represents the standardized value, $x$ represents the observed value of the feature, such as IRI or road distress, $\mu$ represents the mean value for the feature, and $\sigma$ represents the standard deviation of the feature.

### 4.5. Input Data Transformation

In various models, there might be different input formats, thus transforming the data into the correct format for each model is necessary. DNNs are particularly effective for handling organized tabular data arranged in rows and columns. On the other hand, GCNs are specifically designed to process data organized in the form of graphs. In a graph, nodes symbolize entities, edges denote relationships, and both nodes and edges can have associated features. The details of input data for each model are described below:

DNN input data are arranged in a tabular format, where each row corresponds to one road segment and each column corresponds to a feature. Therefore, these data for DNN models can be used in the existing tabular format after creating a format with three-year rolling window formats.

In graph theory, nodes typically represent target objects, while edges indicate the relationships between them. For road network selection, where the target object is a road, using a dual representation that treats roads as nodes is likely to give better results. In addition, GCNs primarily aggregate node features rather than edge features. Therefore, in the current study, the road network was simplified into a dual graph, as shown in Figure 5, while the features of road network selection were reframed as node features in the GCN model [33].

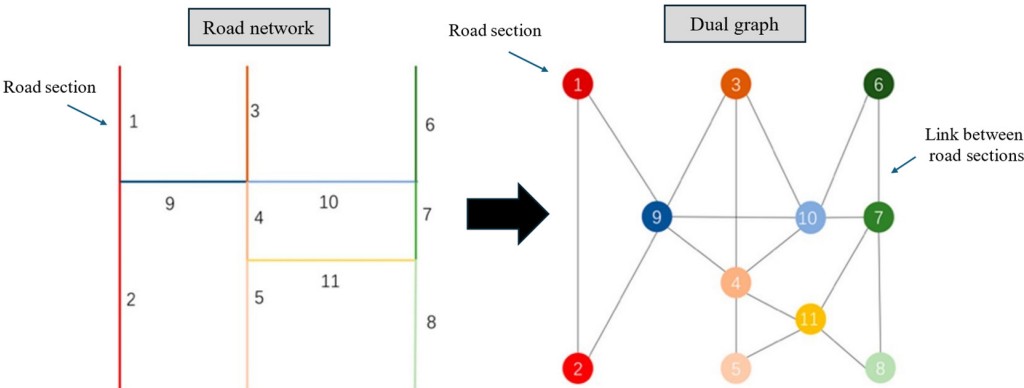

**Figure 5.** Input data for GCN model.

## 5. Model Development

In this section, the DNN was developed to predict the IRI in the highway sections. The contents of this section cover the DNN model used in this research, model performance measurement, and hyperparameter optimization.

Deep learning was applied to predict the IRI values using DNN and GCN models, utilizing the data related to the surface conditions of the national highways in Thailand from 2019 to 2021. As the model was based on supervised learning, the IRI values from 2022 were used as labels. All input data were processed into a format suitable for training the DNN and GCN models. Finally, the prediction performance of the IRI values obtained from both deep learning models was compared across the entire network and in selected provinces. The detailed steps of the procedure are as follows.

### 5.1. Deep Neural Network

The DNN was fully connected and feedforward in type. Models of the DNN were developed as basic mathematical models defining a function $\theta : Z \rightarrow \hat{Y}$, or a distribution over $Z$, or even both $Z$ and $\hat{Y}$. Here, $Z$ represents the set of contextual factors and $\hat{Y}$ represents the future IRI value to be predicted. Assuming $z$ is the daily vector of contextual factors, $z \in Z$, the neuron network function $f(z)$ can be described as a combination of function $g(z)$, while the function of each layer can be broken down into other functions. According to the definition of the neural network function, it is simple to design a network structure with arrows showing the connections between functions. The nonlinear weighted sum is a function that is often used, as shown in Equation (3):

$$f(z) = K(\sum_i w_i g_i(z)) \tag{3}$$

where $w_i$ represents the weight parameter of $g_i$, $g_i$ denotes a specific function within the set $g = g(g_1, g_2, \ldots, g_i)$, and $K$ is a predetermined function, commonly known as the activation function, such as sigmoid, softmax, or rectifier (ReLU). An essential characteristic of the activation function is its ability to provide a smooth transition, as input values vary. The current study used the ReLU function $K(z) = \max(0, z)$ for $K$, which was capable of generating nonlinear values and maintaining non-negativity. The architecture of the DNN models in this research is shown in Figure 6.

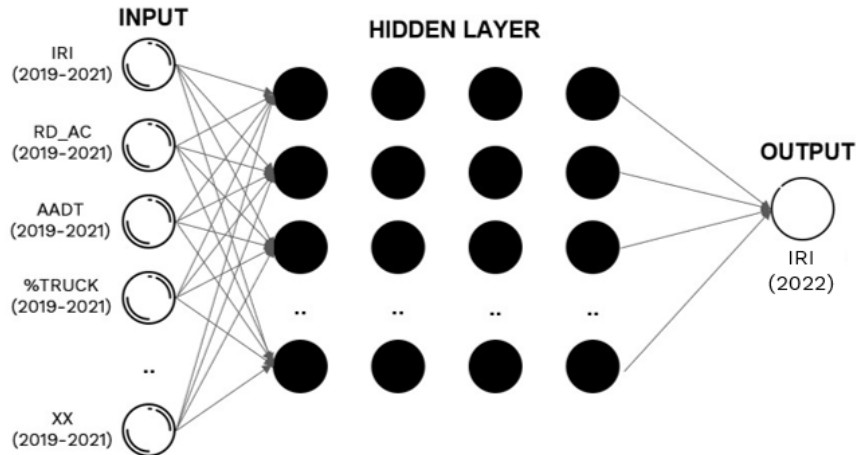

**Figure 6.** Architecture of DNN model.

Given input nodes $(z_i \in Z)$ and only one output node, the predictor, which can be defined as $\hat{y} \in Y$, is shown in Equation (4):

$$\hat{y} = z \cdot W + b \tag{4}$$

where $z$ represents the contextual input factor, $z \in R^d = Z$, $W$ denotes the weight parameter, and $b$ signifies the bias term. If $w$ and $b$ are assigned distinct interpretations and ranges, the formula can be extended to represent the entire neural network, individual layers, or even each neuron within the network. In this approach, a group of units computes a weighted sum based on inputs from the previous layer, which is then passed through a nonlinear function. Upon feeding the contextual factor vector into the network, the internal state (activation) of neurons and layers adjusts in response to the input, ultimately resulting in predictions based on these factors and activation functions. The network is structured by linking the output of specific neurons to the input of others, thereby forming a directed, weighted graph. Both the weights and the activation-computing functions can be adjusted through a process called learning, governed by a learning rule [34].

At each node of a neural network, multiple nonlinear regressions were applied. Within a single layer, the inputs for each node are formed through the combination of nodes from the preceding layer. Similarly, for nodes in subsequent layers, inputs are composed of nodes in varying proportions dictated by their respective coefficients. The amalgamation of inputs across different layers is seen as playing a pivotal role in DNNs, with errors substantially mitigated. A clear definition is required of the loss function $L$ before commencing training of the forecasting model and determining the values for $W$ and $b$. The loss function is crucial in machine learning because it informs how much the solution differs from the best possible solution for the problem to be considered. This function can be written as shown in Equation (5):

$$L(y, \hat{y}) = \frac{1}{2}\|y - \hat{y}\|^2 = \frac{1}{2}\|y - f(z)\|^2 \tag{5}$$

where $y$ represents the actual value, $\hat{y}$ stands for the predicted value, and $z$ denotes a vector of contextual factors across a given input training set. Backpropagation is a technique used to compute the gradient of the loss function (which signifies the associated cost of a particular state) concerning the weights of DNN. The computational expenses incurred during the backward pass (BP) are essentially the same as those of the forward pass (FP). The process involves iteratively conducting both forward and backward passes until an acceptable level of performance is achieved [35]. The weight updates through backpropagation can be accomplished using stochastic gradient descent, as shown in Equation (6):

$$w_{ij}(t+1) = w_{ij}(t) + \eta \frac{\partial L}{\partial w_{ij}} + \xi(t) \tag{6}$$

where $\eta$ represents the learning rate, and $\xi(t)$ stands for a stochastic term. The selection of the loss function is influenced by various factors, such as the learning approach and activation function. Simultaneously, this process can also be utilized to update the parameter $b$.

### 5.2. Graph Convolutional Network

GCNs are semi-supervised models that can process graph structures. They represent a development from CNNs in the field of graphs. GCNs have achieved major advancements in various applications, including image classification [36], document classification [37], and unsupervised learning [38]. The convolutional methods in GCNs involve spectrum and spatial domain convolution [36]. The current study applied the former method. Spectral convolution is defined as the outcome of the product between the signal x on the graph and the filter $g_\theta(L)$, which is constructed within the Fourier domain of $g_\theta(L) * x = U_{g\theta}(U^T x)$, where $\theta$ represents a model parameter, $L$ stands for the graph Laplacian matrix, $U$ denotes the eigenvector of the normalized Laplacian matrix $L = I_N - D^{-\frac{1}{2}} A D^{-\frac{1}{2}} = U\lambda U^T$, and $U^T x$ signifies the graph Fourier transformation of $x$. Additionally, $x$ can be elevated as $X \in R^{N \times C}$, where $C$ denotes the number of features.

With the characteristic matrix $X$ and adjacent matrix $A$, GCNs can substitute the convolutional operation found in previous CNNs by executing the spectrum convolutional operation. This operation considers the graph node and the adjacent domains of nodes

to grasp the spatial traits of the graph. Furthermore, a hierarchical propagation rule is employed to overlay multiple networks. A multilayer GCN model as described in [38] can be represented as shown in Equation (7):

$$H^{(l+1)} = \sigma(\widetilde{D}^{-\frac{1}{2}} \hat{A} \widetilde{D}^{-\frac{1}{2}} H^{(l)} \theta^{(l)}) \tag{7}$$

where $\widetilde{A} = A + I_N$ represents an adjacency matrix with self-connection structures, $I_N$ is an identity matrix, $\widetilde{D}$ is a degree matrix, $\widetilde{D} = \sum_j \widetilde{A}_{ij}$, $H^{(l)} \in R^{N \times l}$ denotes the output of layer $l$, $\theta^{(l)}$ stands for the parameter of layer $l$, and $\sigma(\cdot)$ represents an activation function used for nonlinear modeling. Typically, a two-layer GCN model, as outlined in [38], can be presented as shown in Equation (8):

$$f(X, A) = \sigma(\hat{A} \, ReLU \, (\hat{A} X W_0) W_1) \tag{8}$$

where $X$ serves as a feature matrix; $A$ represents the adjacency matrix; and $\hat{A} = \widetilde{D}^{-\frac{1}{2}} \widetilde{A} \widetilde{D}^{-\frac{1}{2}}$ constitutes a preprocessing step, where $\widetilde{A} = A + I_N$ stands as the adjacency matrix of graph G with a self-connection structure. $W_0 \in R^{P \times H}$ stands for the weight matrix from the input layer to the hidden unit layer, where $P$ denotes the time length and $H$ indicates the number of hidden units. $W_1 \in R^{H \times T}$ denotes the weight matrix from the hidden layer to the output layer. $f(X, A) \in R^{N \times T}$ indicates the output with a forecasting length of $T$, and $ReLU()$ stands as a common nonlinear activation function.

GCNs can be used to encode both the topological structures of road networks and the attributes of road sections concurrently. This is achieved by discerning the topological relationship between the central road section and its surrounding road sections, thereby capturing spatial dependence. In summary, this study acquired an understanding of spatial dependence through the implementation of the GCN model [38]. The architecture of GCN in this study is shown in Figure 7.

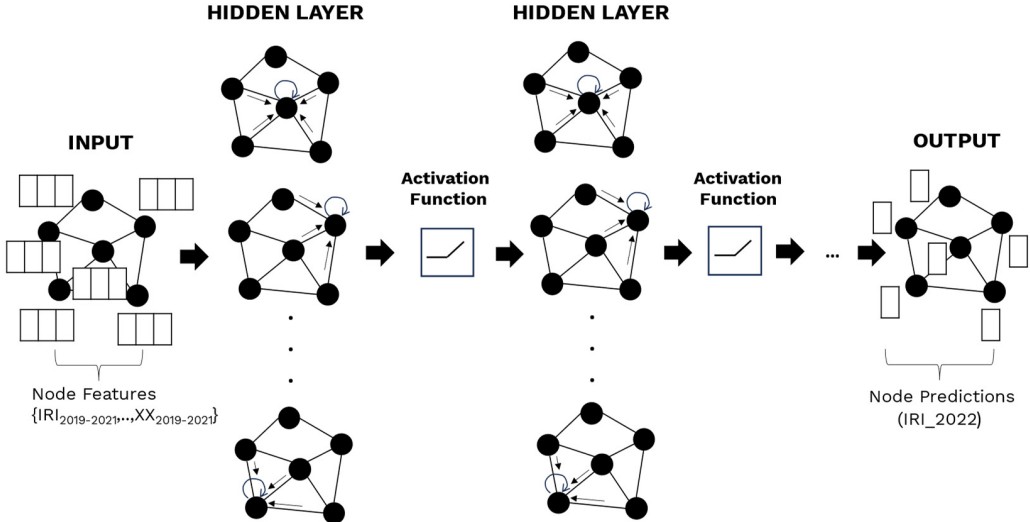

**Figure 7.** Architecture of GCN model.

### 5.3. Importance of Features

Shapley Additive Explanation (SHAP), introduced by Lundberg and Lee [39], has gained popularity as a method for interpreting machine learning model predictions. SHAP is grounded in cooperative game theory [40], where the contributions of individual players (features) to the overall outcome (predictions) are evaluated. The method assigns each

feature a Shapley value, which represents the average marginal contribution of that feature across all possible combinations of features.

The underlying principle of SHAP feature importance is straightforward: features with larger absolute Shapley values are considered more important for the model's predictions. To determine global feature importance, the average absolute Shapley values for each feature across the dataset are calculated using the formula as shown in Equation (9) [41].

$$I_j = \frac{1}{n} \sum_{1=i}^{n} \left| \phi(i)_j \right| \tag{9}$$

where $I_j$ is the average of SHAP values $j$, $n$ is the total number of samples, and $\phi(i)_j$ represents the SHAP value for feature $j$ in sample $i$.

After these importance scores are computed, the features are sorted in descending order of importance, and the results are visualized.

### 5.4. K-Fold Cross-Validation

K-fold cross-validation is a widely adopted technique for evaluating the performance of predictive models. In this method, the dataset is partitioned into k equal subsets or folds. The model is trained on k-1 folds and validated on the remaining fold. This process repeats k times, with each fold serving as the validation set once, while the other k-1 folds are used for training. The final model performance is then computed as the average across all k iterations, yielding a robust estimate of the model's accuracy and generalizability. In this study, 5-fold cross-validation was chosen to balance bias and variance, as empirical research has shown that 5-fold cross-validation often yields stable performance estimates with moderate computational cost [42,43].

### 5.5. Model Performance Measurement

The performance evaluation of each prediction method was based on the following common indicators: the mean absolute percentage error (MAPE), the mean absolute error (MAE), and the coefficient of determination ($R^2$). MAPE indicates the relative accuracy and percentage deviation of estimations, whereas MAE provides the average magnitude of errors, regardless of their directions. $R^2$ represents the proportion of the variance in the target that is explained by the features in the model, with higher values indicating a better fit. The three metrics are defined in Equations (10)–(12), respectively, as follows:

$$MAPE = \frac{1}{n} \sum_{i=1}^{n} \left| \frac{y_i - \hat{y}_i}{y_i} \right| x100 \tag{10}$$

$$MAE = \frac{1}{n} \sum_{i=1}^{n} |y_i - \hat{y}_i| \tag{11}$$

$$R^2 = 1 - \frac{\sum_{i}^{n} (y_i - \hat{y}_i)^2}{\sum_{i}^{n} (y_i - \overline{y})^2} \tag{12}$$

where $n$ is the number of testing samples, $y_i$ is an observed IRI value, and $\hat{y}_i$ indicates a predicted IRI value output by the predicted method.

The standard interpretation of MAPE for IRI prediction models is described in [44]. In the context of this matter, MAPE values of less than 10 percent are categorized as "highly accurate", while MAPE values ranging between 10 and 20 percent are characterized as "good". Additionally, if the MAPE values fall within the range of 20–50 percent, they are classified as "reasonable", and any value below 50 percent is designated as "inaccurate".

### 5.6. Hyperparameter Calibration and Optimization

This section describes hyperparameter tuning and optimization. Five parameters required optimization in the DNN and GCN models: number of hidden layers, number

of hidden nodes, learning rate, epoch, and batch size. The optimal hyperparameters were determined based on a repeated trial-and-error approach and selecting the hyperparameters that resulted in the greatest model performance. These chosen hyperparameters were used in the final DNN and GCN model and are presented in Table 2.

**Table 2.** Hyperparameter values following tuning and optimization.

| Hyperparameter | Value |
|---|---|
| Number of hidden layers | 2 |
| Number of hidden nodes | 100 |
| Learning rate | 0.001 |
| Epoch | 30 |
| Batch size | 64 |

## 6. Results

The feature ranking for the DNN model was analyzed in the model development section. Figure 8 presents the top 10 most relevant features along with their respective importance. Notably, IRI values from previous years played a crucial role in IRI prediction, especially those from the previous two years. The IRI from the previous year was of less importance than that from two years prior due to a lower data volume in that year, likely caused by COVID-19, followed by the influence of road hierarchy and temperature.

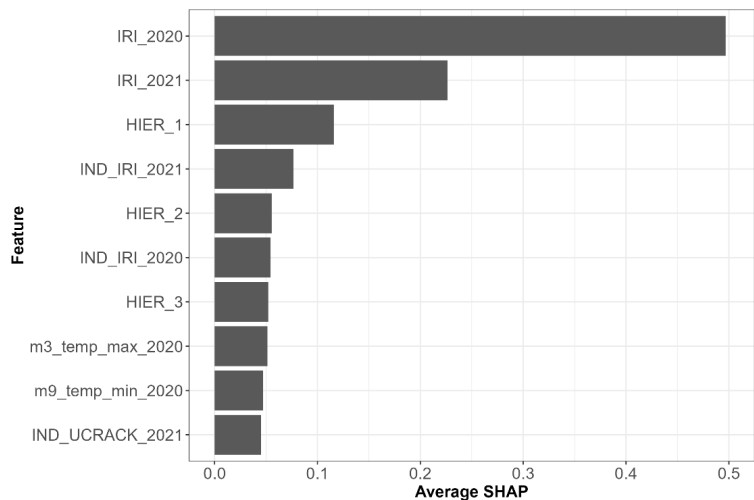

**Figure 8.** Most relevant features and their importance for IRI prediction using the SHAP method.

The model results were divided into two types: performance across the entire network, using data from the full network, and performance within specific areas, selected from provinces with varying continuous spatial data as identified by the beta index ($\beta$).

### 6.1. Comprehensive Network

Table 3 compares the performance of the DNN, GCN, and multiple linear regression (MLR) models, using the same features as the DNN and GCN, based on three common metrics (MAPE, MAE, and $R^2$) for the entire network.

**Table 3.** Performance of IRI prediction models in entire network.

| Model | MAPE (%) | MAE (m/km) | $R^2$ |
|---|---|---|---|
| DNN | 15.12 | 0.414 | 0.665 |
| GCN | 15.23 | 0.432 | 0.682 |
| MLR | 17.14 | 0.445 | 0.641 |

Based on the results from all models for the entire network of IRI data, the DNN and GCN models outperformed the traditional model, MLR. The DNN model had values for MAPE, MAE, and $R^2$ of 15.12%, 0.414 m/km, and 0.665, respectively, while the GCN model had values of 15.23%, 0.432 m/km, and 0.682, respectively.

The performance levels of both models were similar, with the DNN option producing slightly better performance for all parameters. This was likely because the IRI data consisting of the entire network, links, and nodes, was not highly continuous. When combined on a large scale across the entire network, the DNN model produced results that were comparable to those for the GCN model.

### 6.2. Specific Areas

From the test dataset, specific areas were selected to assess the model's performance. The selection was based on provinces that consist of different connectivity ratios. The measure of graph connectivity is given by the beta index (β), which measures the density of connections and is defined as shown in Equation (13) [45].

$$\beta = \frac{E}{V} \tag{13}$$

where $E$ is the total number of edges (links between road sections) and $V$ is the total number of nodes (road sections) in the network. An example of beta index calculation is shown in Figure 9.

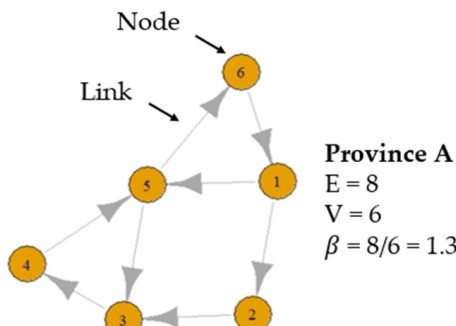

**Figure 9.** Example of beta index calculations for each province.

The levels of the beta index are categorized by the 25th percentile and 75th percentile of all provinces' beta indexes in the dataset, resulting in the following classifications: low index (less than 0.64), medium index (between 0.64 and 0.67), and high index (greater than 0.67). The specific areas were shown by the color-filled polygons in Figure 10.

Summarizes the performance of the DNN, GCN, and MLR based on three common metrics (MAPE, MAE, and $R^2$) in specific areas. It was found that, in certain provinces, the DNN outperformed the GCN, while in other provinces, the GCN demonstrated superior performance compared to the DNN. Notably, most of the DNN and GCN models consistently outperformed the MLR model across all provinces and are presented in Table 4.

Figure 11 compares the beta index, representing network connectivity, with the MAPE difference between DNN and GCN models. Positive values on the y-axis indicate better performance by GCN, while negative values favor DNN. Contrary to expectations, no clear trend shows GCN outperforming DNN as the beta index increases from 0.57 to 0.75. Although higher connectivity was expected to enhance GCN's ability to capture spatial dependencies, the MAPE difference fluctuates across beta index values with no consistent pattern.

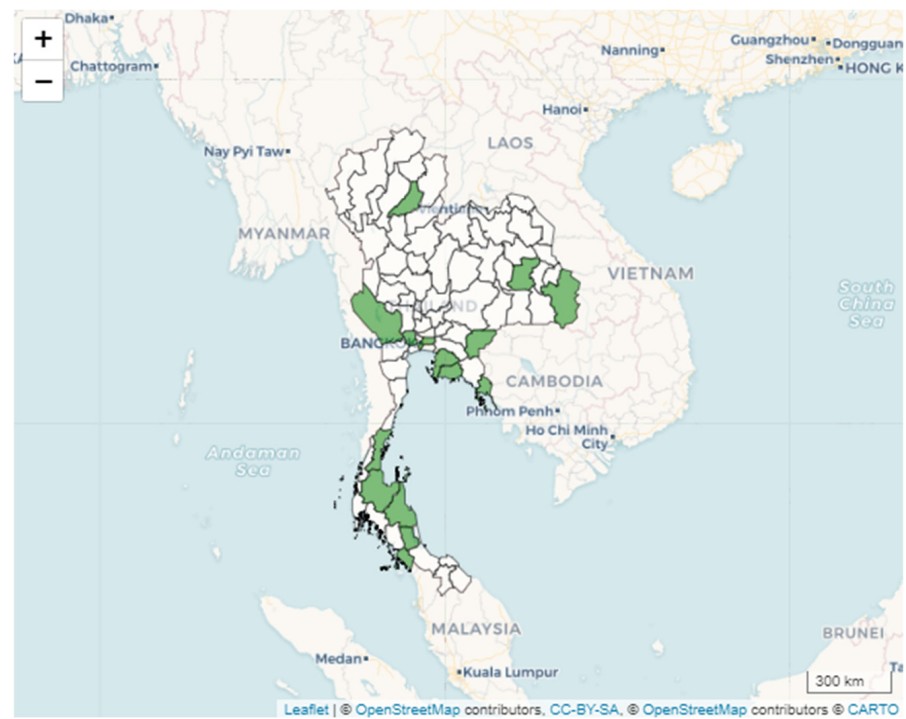

**Figure 10.** The geographic locations of the selected provinces in the test dataset.

**Table 4.** Performance of IRI prediction models across provinces in the test dataset.

| No. | Province Name | Beta Index | DNN | | | GCN | | | MLR | | |
|---|---|---|---|---|---|---|---|---|---|---|---|
| | | | MAPE (%) | MAE (m/km) | $R^2$ | MAPE (%) | MAE (m/km) | $R^2$ | MAPE (%) | MAE (m/km) | $R^2$ |
| 1 | Bangkok | 0.63 | 12.75 | 0.449 | 0.648 | 13.84 | 0.493 | 0.653 | 13.87 | 0.493 | 0.592 |
| 2 | Chon Buri | 0.61 | 15.15 | 0.476 | 0.638 | 18.25 | 0.621 | 0.495 | 17.67 | 0.522 | 0.626 |
| 3 | Rayong | 0.66 | 17.08 | 0.398 | 0.776 | 16.46 | 0.409 | 0.793 | 24.26 | 0.492 | 0.735 |
| 4 | Trat | 0.67 | 14.34 | 0.467 | 0.476 | 19.41 | 0.602 | 0.242 | 16.61 | 0.473 | 0.579 |
| 5 | Sa Kaeo | 0.75 | 14.32 | 0.412 | 0.428 | 15.48 | 0.459 | 0.157 | 15.64 | 0.421 | 0.465 |
| 6 | Ubon Ratchathani | 0.68 | 14.74 | 0.378 | 0.778 | 13.04 | 0.357 | 0.796 | 17.68 | 0.434 | 0.757 |
| 7 | Roi Et | 0.69 | 16.34 | 0.393 | 0.765 | 13.84 | 0.371 | 0.739 | 17.79 | 0.419 | 0.733 |
| 8 | Phrae | 0.63 | 12.84 | 0.308 | 0.825 | 9.34 | 0.228 | 0.820 | 13.43 | 0.305 | 0.826 |
| 9 | Kanchanaburi | 0.62 | 15.14 | 0.370 | 0.509 | 14.15 | 0.366 | 0.389 | 16.65 | 0.396 | 0.470 |
| 10 | Nakhon Pathom | 0.64 | 12.50 | 0.288 | 0.764 | 12.20 | 0.298 | 0.544 | 13.59 | 0.294 | 0.732 |
| 11 | Nakhon Si Thammarat | 0.68 | 17.62 | 0.465 | 0.599 | 16.58 | 0.449 | 0.684 | 18.79 | 0.447 | 0.686 |
| 12 | Surat Thani | 0.69 | 16.13 | 0.409 | 0.589 | 18.63 | 0.477 | 0.533 | 17.44 | 0.415 | 0.632 |
| 13 | Chumphon | 0.67 | 14.09 | 0.485 | 0.504 | 14.96 | 0.517 | 0.454 | 14.21 | 0.458 | 0.587 |
| 14 | Satun | 0.69 | 12.05 | 0.357 | 0.358 | 13.16 | 0.411 | 0.069 | 14.33 | 0.377 | 0.465 |
| 15 | Phatthalung | 0.67 | 14.51 | 0.496 | 0.445 | 17.17 | 0.544 | 0.353 | 14.59 | 0.448 | 0.599 |

Remark: gray shading represents the models with lowest MAPEs.

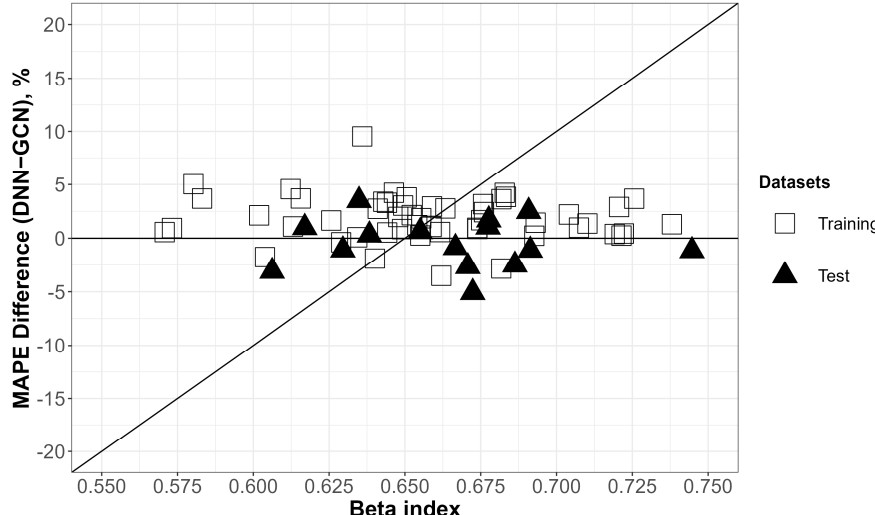

**Figure 11.** Scatter plot between the beta index and the difference in MAPE between the DNN and GCN, where one data point represents data from one province.

This variability may stem from the dataset's limited beta index range (0.57 to 0.75) and the predominantly linear structure of the national highways. Figure 12 highlights that intercity highways generally exhibit low connectivity, with beta indices around 0.75. Additionally, gaps in IRI data due to non-annual data collection further reduce effective connectivity, limiting GCN's advantage.

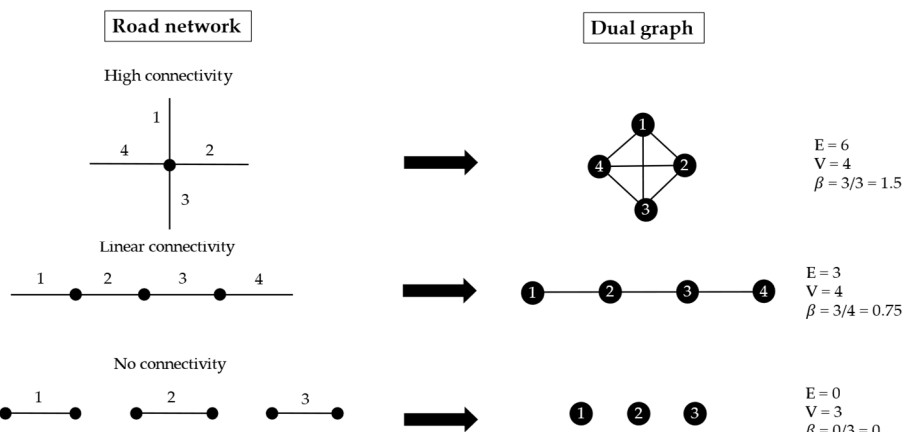

**Figure 12.** Various road network types with different beta indices.

These constraints suggest that DNN performs better for low-connectivity road networks, while GCN might be more effective in highly connected networks, such as urban systems. Thus, for linear or sparsely connected highway networks like those in this study, DNN remains a suitable choice, with GCN's potential benefits becoming pronounced only in denser networks.

## 7. Discussion and Conclusions

Advanced deep neural networks are increasingly used in traffic prediction, yet few studies have integrated spatial and temporal dependencies for IRI modeling. This study addresses this gap by applying deep learning to predict IRI values using DNN and GCN models, utilizing surface condition data from Thailand's national highways (2019–2021). IRI values from 2022 served as labels, with all input data processed for training the models based on supervised learning.

The DNN and GCN models were trained using a trial-and-error approach to optimize their hyperparameters (hidden layers, hidden nodes, learning rate, epochs, and batch size) for the best performance. Key features influencing the prediction of the IRI included the previous year's IRI, highway hierarchy, and climate factors. The previous year's IRI was the most significant, highlighting the importance of historical data in modeling. Highway hierarchy affected roughness due to varying traffic loads, while climate factors like temperature influenced pavement degradation.

Each model's performance was evaluated using MAPE, MAE, and $R^2$ was compared with the traditional MLR model. The research results were categorized into two types: model performance across the entire network and model performance within specific areas. In analyzing the comprehensive network, both the DNN and GCN models outperformed the MLR model, likely due to their ability to capture complex, nonlinear patterns in the data that the linear MLR model could not. The DNN and GCN models showed similar performance across the entire network, indicating that both can generalize well when applied to a wide-ranging dataset. DNN and GCN each outperformed the other across different beta index levels, suggesting that the beta index in this study may not be high enough to reveal a clear distinction between the two models. This outcome is likely influenced by the structure of the national highway network. However, the results indicate that DNN alone was effective for road networks with a beta index below 0.74. Future research should apply the GCN to road networks with higher connectivity levels, where the model's strengths in spatial representation may lead to more pronounced performance gains over the DNN, thus enabling a clearer distinction between the capabilities of the GCN and DNN models in road network analysis. Moreover, as previous studies [15,16] have shown, the GCN generally performs better in predicting traffic data. It is also important to consider other factors, such as the connectivity index.

Nevertheless, data limitations included the aggregation of data over the entire year, with some data potentially collected at different times throughout the year. Another limitation was that certain IRI values in some years decreased despite no road maintenance occurring, or increased even after maintenance, due to IRI measurements either not being collected immediately after road maintenance, not being recorded in the database, or being recorded in a different database that was not used in this study. To improve the accuracy of IRI predictions, future researchers or agencies should collect data in a more detailed and systematic way. For example, gathering IRI data immediately after road maintenance and at regular intervals afterward (e.g., semiannually or quarterly) would help track road quality changes over time and provide a better understanding of road deterioration. This approach would make the data easier to analyze and more useful for predicting road surface conditions effectively.

**Author Contributions:** The authors confirm their contributions to the paper as follows. S.B.: supervision, conceptualization, methodology, and writing—review and editing; C.A.: formal analysis, validation, visualization, and writing—original draft preparation; K.J.: conceptualization, investigation, data interpretation, and writing—review and editing; P.L.: project administration, project oversight, resources and funding acquisition; A.S.: acquisition of data, resources, and provision of critical feedback. All authors have read and agreed to the published version of the manuscript.

**Funding:** This research was supported by the Research and Development Division in the Department of Highways, Thailand, through the Highway Pavement Life Cycle Analysis Project (Project No. 651107000020).

**Institutional Review Board Statement:** Not applicable.

**Informed Consent Statement:** Not applicable.

**Data Availability Statement:** The datasets presented in this article are not readily available because the data belong to the Department of Highways and are not publicly accessible. Requests to access the datasets should be directed to the Department of Highways, Thailand.

**Acknowledgments:** Data used in this study were from the Thailand Pavement Management System (TPMS), Bureau of Highways Maintenance Management in the Department of Highways, Thailand.

**Conflicts of Interest:** Author Chuthathip Athan was employed by Mobinary Company Limited,. The other authors declare that the research was conducted without any commercial or financial relationships that could be perceived as potential conflicts of interest.

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
