# Peer review of "Comparative Analysis of Deep Neural Networks and Graph Convolutional Networks for Road Surface Condition Prediction"

_sustainability, doi:10.3390/su16229805_

Round 1

Reviewer 1 Report

Comments and Suggestions for Authors

The primary question addressed by the research is how deep neural networks (DNNs) and graph convolutional networks (GCNs) compare in predicting international roughness index (IRI) values for evaluating road surface conditions. The study also investigates the effect of incorporating additional predictor features (e.g., type and timing of road work) on model performance.

The topic is original and highly relevant to infrastructure maintenance, particularly in road safety and predictive maintenance. Road surface condition monitoring is a critical task for highway agencies, and optimizing it can result in significant cost savings and better road safety outcomes. The comparison between DNNs and GCNs for IRI prediction, particularly the integration of spatial data, addresses a specific gap in the field of predictive road condition modeling. This is particularly important given the increasing use of data-driven methods for infrastructure management.

This study contributes valuable insights by directly comparing two advanced machine learning models—DNNs and GCNs—on their effectiveness for road condition prediction. While DNNs are commonly used in large-scale data applications, the introduction of GCNs offers a more tailored approach for highway networks due to their inherent graph-like structure. Additionally, the study introduces a novel feature by integrating recent road work details, which could significantly influence future IRI values but is often overlooked in prior research.

Improvements to Methodology and Controls:

·         Data Granularity: The authors should provide more information about the granularity and consistency of the dataset. For example, is the road work data uniformly distributed across the network, or does it vary regionally? Understanding how the data is spread may help explain variations in the model's performance.

·         Cross-Validation and Overfitting: Additional cross-validation techniques could ensure that both models are not overfitting to particular data regions. Considering that GCNs performed better in areas with continuous data, was there any particular emphasis on avoiding overfitting in such regions?

·         Model Interpretability: The inclusion of more interpretability methods such as SHAP (Shapley Additive Explanations) could help validate why certain features, like the timing of road work, had more or less influence on the predicted IRI values.

·         Feature Importance Analysis: A deeper analysis of which predictor features (e.g., weather, traffic volume, road work) had the most impact on model predictions would be helpful. Knowing which features contribute most to model accuracy can guide data collection efforts in the future.

The conclusions are consistent with the presented evidence. The research identifies the contexts where each model performs best. DNNs are more effective with large, non-continuous datasets, whereas GCNs excel in continuous spatial data environments. This distinction is a crucial takeaway for highway agencies, as it can inform which model to deploy based on the nature of the available data.

The references appear appropriate, but the authors should ensure that they include recent studies (within the last 3–5 years) that explore machine learning in infrastructure or road maintenance applications. As the field of predictive modeling is rapidly evolving, having a strong foundation of current literature would be beneficial.

Reviewer 2 Report

Comments and Suggestions for Authors

This study describes a comparative analysis of deep neural networks and graph convolutional networks in road condition prediction, but there are still some issues that need to be addressed.

1. The introduction section mentions that previous research was mainly based on linear or nonlinear regression techniques, but there seems to be no examples provided in the article to illustrate this aspect.

2. It is not recommended to cite references in the title of Figure 4.

3. The fifth chapter mainly introduces two types of models, and it is not recommended to include section 5.4 on hyperparameter settings for the models.

4. The comparative experiment between the prediction results of the two models in the article is not sufficient. It is suggested to supplement the comparative experiment.

5. Reference chapters should not have numbering.

6. Specific explanations should be provided for all character variables mentioned in the text.

7. Most of the references cited in the article are not from the past three years. It is recommended to update the references.

Reviewer 3 Report

Comments and Suggestions for Authors

This article compares two models for predicting road conditions, the study is well performed and the data supports the conclusion. Recommend for publishing. 

Reviewer 4 Report

Comments and Suggestions for Authors

This paper is logical and well-organized. The main comments are as follows:

1The article uses two ways (a transverse profile logger and the laser crack measurement system) to obtain IRI data. For the same test section, are there any differences in the data obtained in these two ways? This affects the accuracy of the final prediction results.

2It is recommended to supplement Figure 2 with data labels, such as the maximum, minimum, and median values of the box plots, which will make it easier to understand the information about the IRI in different roads.

3Suggest adding a scale bar in Figure 9.

4In section 6.2, the R2 of the two models are 0.670 and 0.786 respectively, which shows that the models are not well applied from the data results.

5Suggest enhancing the clarity of images, such as Figures 4, 7, 8, 10, etc.

Reviewer 5 Report

Comments and Suggestions for Authors

Thank you so much for the review invitation for "Comparative Analysis of Deep Neural Networks and Graph Convolutional Networks for Road Surface Condition Prediction" by Boonsiripant et al.. Here are the major and minor comments:

Major comments:

  • Given that you selected one of the provinces for training and testing for Table 4, can you use the model to predict the rest of the provinces as external validation? Then, essentially performing out-of-sample generalization. This will provide a better understanding of the useability and generalizability of the model trained on high beta provinces, and this might help readers to understand if they should only model differently depending on beta values. Currently, we only see data improvement for GCN in one of the 77 provinces.

Minor comments:

  • For Figure 3, while it is okay to use numeric values for the plot, it is necessary to include the key in the legend even if it is mentioned in the text. This is to ensure the figure is self-contained. There is a sharp decline of 22100, is there a reason? Would this impact the IRI evaluation?
  • In table 1, for the data attributes, why are the "Binary" or "binary" binary? Binary implies they are 0 or 1, but these are numeric variables. 

Round 2

Reviewer 2 Report

Comments and Suggestions for Authors

This revised research paper is almost free of major issues, but it is advisable to verify the correctness of formula 10 before the article is officially published.

Reviewer 4 Report

Comments and Suggestions for Authors

The author has made the necessary modifications as requested. Additionally, the scale bar in Figure 10 has been hidden in the PDF file. Please check.

After completing the revisions, the paper can be accepted.

Reviewer 5 Report

Comments and Suggestions for Authors

Thank you for addressing all my comments. I have no further questions. 

Author Response

Thank you very much.